# Family Medicine Academic Workforce of Medical Schools in Taiwan: A Nationwide Survey

**DOI:** 10.3390/ijerph18137182

**Published:** 2021-07-05

**Authors:** Shu-Han Chen, Hsiao-Ting Chang, Ming-Hwai Lin, Tzeng-Ji Chen, Shinn-Jang Hwang, Ming-Nan Lin

**Affiliations:** 1Department of Family Medicine, Taipei Veterans General Hospital, Taipei 11217, Taiwan; u9601705@cmu.edu.tw (S.-H.C.); htchang.tw@gmail.com (H.-T.C.); mhlin@vghtpe.gov.tw (M.-H.L.); sjhwang@vghtpe.gov.tw (S.-J.H.); 2Faculty of Medicine, School of Medicine, National Yang Ming Chiao Tung University, Taipei 11217, Taiwan; 3Taiwan Association of Family Medicine, Taipei 10046, Taiwan; mingnan.lin@gmail.com; 4Department of Family Medicine, Dalin Tzu Chi Hospital, Chiayi County 62247, Taiwan

**Keywords:** family medicine, faculty development, medical education, public administration, public policy, healthcare, primary care

## Abstract

Little is known about family medicine academic workforce in Taiwan, and basic data on this may aid healthcare decision-makers and contribute to the limited literature. We analyzed data from 13 medical schools in Taiwan collected by the Taiwan Association of Family Medicine from June to September 2019, regarding characteristics of medical schools, and total staff, gender, age, degree, working title (adjunct/full-time), academic level, and subspecialty of each current family medicine faculty member. Total 13 medical schools in Taiwan with an undergraduate education program in family medicine, but only nine of the 13 medical schools had family medicine departments, while four still do not. A total of 116 family medicine faculty members ranging from 33–69 years. Of these, most were male (*n* = 85, 73.3%), with a mean age of 43.3 years. Most faculty members possessed a master’s degree (*n* = 49, 42.2%), were academic lecturers (*n* = 49, 42.2%), were located in northern Taiwan (*n* = 79, 68.1%), and subspecialize in gerontology and geriatrics (*n* = 55, 47.4%) and hospice palliative care (*n* = 53, 45.7%). Additionally, most family medicine faculty in medical schools were adjunct faculty (*n* = 90, 77.6%), with only about one-fourth (*n* = 26, 22.4%) working full-time. Our study provides the most holistic census to date on academic family medicine faculty from all medical schools in Taiwan. The novel information can provide educational leaders, health policy managers, and decision-makers about the current developments of the family medicine departments in Taiwan’s medical schools. The basic data will help formulate an effective medical school family medicine education plan and improve the establishment and development of the family medicine faculty workforce to help medical education and national health policy development in the future in Taiwan.

## 1. Introduction

The family medicine workforce plays an increasingly important role as healthcare providers, leaders, managers, supervisors, and overall coordinators of personal and community healthcare, especially since the beginning of the COVID-19 pandemic [1]. Family medicine, also called general practice (GP) or family practice, is the medical specialty effector of primary healthcare (PHC). It has its own body of knowledge, with a functional unit consisting of the family and the individual, is based on clinical, epidemiological, and social methods, and integrates the biological, clinical, and behavioral sciences [2]. Since 2008, the World Health Organization has listed family medicine as the discipline most closely related to this type of healthcare and renewed the call for the development of high-quality primary care worldwide [3]. The American Council of Family Medicine defined family practice as a medical specialty that cares about the care of the total health of the individual and the family, and integrates the clinical, biological, and behavioral sciences. Its scope is not limited by age, sex, organ, system, or morbid entity [2].

Generally, in international medical practice, the term family doctor is used to refer to doctors whose basic function is to serve the community, the family, and the individual in a determined health area, regardless of whether or not they have specialized postgraduate training. This is a physician responsible for the general care of the patient in a hospital or clinical setting. As we know, family medicine is a three-dimensional profession that combines knowledge, skills, and processes. Although knowledge and skills may be shared with other professions, the family medicine process is unique. A family medicine physician may also supervise and teach medical students, interns, and residents who are involved in patient care. 

The family medicine profession began in the 1960s and spread to various countries in the 1970s [4]. In the period between the French and the Industrial Revolutions, the term ‘family doctor’ (médico de cabecera) first appeared, referring to a doctor standing by the patient’s bed (cabecera) and providing care in the patient’s home [2]. Therefore, the term is associated with that of general practitioners. Family medicine is a continuation of historical medicine and current performance practitioners. For thousands of years, these generalists have provided all available medical services. They diagnose and treat the diseases, perform surgery, and deliver babies. However, as medical knowledge expands and technology advances, many doctors choose to restrict their practice in specific and defined medical fields. 

In the first half of the 20th century, general medicine began to decline, replaced by medical specialization. Medical schools began to lose interest in general medicine, and the lack of postgraduate training in general medicine led to its decreased intellectual challenge. Subsequently, the international medical community realized that newly graduated doctors were unprepared to address the full range of people’s health problems. As a result, general medical colleges and colleges specializing in the academic development of family medicine have been established in many countries. In the following years, medical schools opened family medicine departments and developed training plans for family medicine faculty. The movement towards establishing family medicine departments in medical schools originated in the United Kingdom in 1953, and the American Medical Association (AMA) in 1990 adopted the principle that all medical schools in the United States must have a family medicine department. 

According to the Constitution of the Republic of China, the first official document advocating primary care as primary infrastructure was promulgated in Taiwan on 1 January 1947. ‘The state, in order to improve national health, shall establish extensive services for sanitation and health protection, and a system of public medical service’. Until the 1970s, the medical system had undergone foundational changes, including the development of various medical specialties. In view of the development of a highly specialized medical system, decentralized medical care is not conducive to the long-term national health of the people of Taiwan. Therefore, the Taiwanese government learned from the experience of the United States and began to promote family medicine [5]. The coronavirus crisis has highlighted many inherent weaknesses in our existing healthcare system [6]. We must pay more attention to the importance of the family medicine workforce. Family medicine providers can help countries worldwide maintain and improve their health and well-being by developing more productive, coordinated, and cost-effective healthcare methods and optimizing the primary healthcare system [7].

A study showed medical students’ opinions about the advantages of family medicine education (e.g., improvement in their communication, history taking, and consultation skills and learning how to manage common health problems in primary healthcare settings) [8]. More countries are also beginning to pay attention to the knowledge level of medical students in family medicine because it is related to the provision of primary healthcare services and the effectiveness of the health service system in the near future after graduation [9]. Consequently, it also influences the primary care physician workforce in the healthcare system. Further, the characteristics of medical schools (e.g., public or private), students’ learning experiences, the medical education programs at medical schools, and family medicine role models, are key factors related to the development of the future family medicine workforce [10]. Above all, we can see the importance of family medicine education. However, according to our literature search, no prior study has shown the status of the academic family medicine workforce of medical schools in Taiwan. Therefore, this study aims to evaluate this to help the family medicine faculty workforce, medical education and training programs, and national health policy development in the future.

## 2. Materials and Methods

### 2.1. Database of Major Medical Schools in Taiwan

We used Taiwan’s Ministry of Education’s public information platform database for colleges and universities to collect data from all of Taiwan’s medical schools and conduct category queries by field. The fields of medical, health, and social welfare are divided into two categories: (1) medical health and social welfare and (2) medical. After searching the Department of Hygiene, a total of eight academic categories were identified (namely, dentistry, medical science, nursing and midwifery, medical technology and laboratory science, treatment and rehabilitation, pharmacy, and traditional medicine). A search was then conducted within the medical category for complementary medical treatment and other medical and health sciences, and a total of 16 public and private universities were identified. Of these 16 universities, 13 general universities (four public and nine private) have medical schools in Taiwan. The four public medical schools are National Taiwan University, National Yang-Ming University (in 2020, it merged with National Chiao Tung University and changed its name to National Yang-Ming Chiao Tung University), National Defense Medical Center, and National Cheng Kung University. The nine private medical schools are Taipei Medical University, Mackay Medical College, Fu Jen Catholic University, Chang Gung University, China Medical University, Chung-Shan Medical University, Kaohsiung Medical University, I-Shou University, and Tzu Chi University [11].

### 2.2. Definitions of Medical Specialties and Subspecialties in Taiwan

In the Diplomate Specialization and Examination Regulations issued by the Ministry of Health and Welfare in 2018, 23 physician specializations included family medicine [12]. However, the subspecialties of geriatrics, hospice and palliative medicine, environmental and occupational medicine, obesity medicine, international travel medicine, adolescent medicine, and osteoporosis medicine are not specialized in the Ministry of Health and Welfare certificate [13].

### 2.3. Defintion of Taiwan Geographical Distribution

The Taiwan area includes the northern, central, southern, and eastern regions. The northern region includes the cities of Taipei, New Taipei, Keelung, Hsinchu, and Taoyuan, and the counties of Hsinchu and Yilan. The central region includes Taichung City and the counties of Miaoli, Changhua, Nantou, and Yunlin. The southern region includes the cities of Kaohsiung, Tainan, and Chiayi, and the counties of Chiayi, Pingtung, and Penghu. The eastern region includes the counties of Hualien and Taitung [11].

### 2.4. Data Analysis of Family Medicine Academic Faculty in Taiwan Medical Schools

Our survey analyzed data collected between June and September 2019 from all 13 medical schools in Taiwan, and the data were collected by the Taiwan Association of Family Medicine. Initially, we used descriptive statistics to analyze the demographic data, including items that clarify the demographics and practice characteristics of the medical schools and family medicine faculty, including the number of current faculty by gender, age, degree, commitment (adjunct/full-time), academic rank, subspecialty, faculty degree, and faculty year. The aim of this study is to clarify the characteristics of medical schools (i.e., public or private, regionally) and the academic family medicine workforce in Taiwan, public or private schools, different regions, and the academic family medicine workforce in Taiwan. Microsoft Excel 2019 software (Microsoft, Redmond, Washington, U.S.) was used to perform data analysis, while the mosaic plot and grouped boxplot package of the R software (version 4.0.5, R foundation, Vienna, Austria) were used to create the mosaic plot (Figure 1) and the boxplot (Figure 2), respectively. Figure 1 illustrates the gender distribution of faculty rank of academic family medicine faculty in Taiwan. Figure 2 illustrates the age and gender distribution of family medicine academic faculty by rank. Furthermore, because of the wide variance in age distribution, median age values are presented by incorporating mean values to the box plot package.

## 3. Results

According to the analysis, the results indicate that not all medical schools have a family medicine department. Among the 13 medical schools in Taiwan, nine (69.2%) (National Taiwan University, National Yang-Ming University, Taipei Medical University, National Defense Medical Center, China Medical University, Chung Shan Medical University, National Cheng Kung University, Kaohsiung Medical University, and Tzu Chi University, listed from north to south and west to east) have a family medicine department. The remaining four (30.8%) (Mackay Medical College, Fu Jen Catholic University, Chang Gung University, I-Shou University) had a family medicine department in an affiliated hospital but not in a medical school (Table 1).

A total of 116 family medicine physicians held the faculty positions. Table 2 summarizes the current distribution of medical faculty in medical schools. Nine medical schools with family medicine departments had a total of 99 medical faculty members, with an average of 11 medical faculty members in the family medicine department of each medical school. In the four medical schools without a family medicine department, there were a total of 17 medical teaching positions, and each school had an average of 4.25 medical faculty positions. Public medical schools have more medical faculty members than private medical schools (54.3% vs. 45.7%, respectively), but the difference was not significant. However, the difference was statistically significant between the northern, central, southern, and eastern regions is statistically significant (68.1%, 12.1%, 11.2%, and 8.6%, respectively).

The characteristics of all 116 faculty members in the Department of Family Medicine are shown in Table 3. There are 2.7 times as many males (*n* = 85, 73.3%) as females (*n* = 31, 26.7%). The age ranged from 30 to 69 years, and the most frequent age category was 40–49 years old (34.5%), followed by 50–59 years old (33.6%), 30–39 years old (19%), and 60–69 years old (12.9%). Lecturers accounted for the majority (42.2%) of faculty, followed by assistant professors (31%), associate professors (16.4%), and professors (10.3%). There were only two family medicine faculty members with more than 20 years of teaching experience, 17.2% had 10–20 years of teaching experience, and the majority (81.1%) had less than 10 years of teaching experience. The most commonly held degree among family medicine faculty is a master’s degree (*n* = 49, 42.2%), followed by a PhD (*n* = 38, 33%), and finally a bachelor’s degree (*n* = 29, 25%). However, all the 116 faculty members possessed an MD degrees. Most medical faculty did not hold full-time positions; only 20.7% of all family medicine faculty are full-time in medical schools (*n* = 24), and the other 79.3% are part-time (*n* = 92, three physicians are full-time in clinics, and 89 physicians are full-time in medical centres, regional hospitals, or regional hospitals). All medical faculty are family medicine specialists; most have subspecialties (*n* = 93, 80.2%), and a few do not (*n* = 23, 19.8%), with geriatrics being the most widely adopted subspecialty (47.4%), followed by hospice palliative medicine (45.7%), environmental and occupational medicine (11.2%), obesity medicine (9.5%), international travel medicine (4.3%), adolescent medicine (2.6%), and osteoporosis medicine (*n* = 2, 1.7%).

The total numbers of full-time staff members in the family medicine departments of Taiwan medical schools comprises 20 males (77%) and 6 females (23%). The number of full-time male faculty positions increased with rank. There were more male full-time professors than associate professors, assistant professors, and full-time lecturers (50%, 35%, 15%, and 0%, respectively). Conversely, there were no full-time female professors. Half of the women are assistant professors, and the remaining female full-time faculty members are associate professors and lecturers. In addition, part-time teaching positions in family medicine are also more frequently held by men than women (*n* = 65 vs. *n* = 25), but contrary to the case for full-time teaching positions, the number of both male and female positions decreases with rank, with lecturers making up the largest proportion of both male and female part-time teaching positions (Table 4).

With increasing rank, the number of family medicine faculty members in Taiwan medical schools gradually decreases for both men and women. In particular, no women are family medicine professors in medical schools; the largest area in the figure indicates that the proportion of male lecturers is the highest (Figure 1).

Figure 2 displays the average sex and age distribution for each rank level. Most faculty members in the departments of family medicine are lecturers (*n* = 49, 42.2%), with an age range between 33 years (youngest) and 65 years (oldest). The average age of male, female, and overall family medicine lecturers is 43.4, 43.1, and 43.3 (±8.09) years, respectively. Similarly, this distribution is 47.7, 42.6, and 46.3 (±6.77) years among assistant professors (n = 36, 31.0%) and 53.4, 52.4, and 53.1 (±6.91) years for associate professors (n = 19, 16.3%), respectively; The average age of male and overall professors (*n* = 12, 10.3%) is 58.1 and 58.1 (±4.89), respectively.

## 4. Discussion

### 4.1. Main Findings and Possible Explanations

Medical schools both shape and are shaped by healthcare systems. A previous Asian Pacific study found 11 medical schools had a family medicine department in Taiwan in 2014 [14]. However, in our study, we found in 2019, there are 13 medical schools in Taiwan with an undergraduate education program in family medicine. Our results indicate that Taiwan established two new medical schools between 2014 and 2019, but only nine of the 13 medical schools have family medicine departments, while four still do not. We found that most medical schools with established family medicine departments are public medical schools in Taiwan, which increased the likelihood of plans for family medicine [10]. Therefore, additional efforts are needed to ensure that all medical schools have a department of family medicine to implement principles and practices of family medicine in the healthcare system in Taiwan. Most private medical schools lacking a family medicine department established a family medicine department in their university hospital. This may be because private medical schools were established after the affiliated hospitals. Medical schools were established to support hospital needs through basic research, medical education to improve limited clinical healthcare services. In the early developmental stages, some private medical schools established educational alliances with other medical schools to share their medical faculty to meet the teaching needs [15].

Our findings suggested that distribution of family medicine academic faculty is consistent with the distribution of medical schools in Taiwan. Moreover, this distribution is also similar to the age and educational level distribution of Taiwan’s civilian population aged 15 years and over in 2019 [7]. However, the regional distribution of family medicine faculty is imbalanced, as most medical schools are located in northern Taiwan, which is consistent with the urban–rural disparity in geographical and temporal availability of medical healthcare services and hospitals [16,17]. Another study demonstrated that family medicine education in rural communities can lead to the development of health service interventions supporting the recruitment and retention of physicians [18]. Therefore, it is important to consider the role of a geographically balanced distribution of family medicine faculty in Taiwan, while formulating future policies.

Research indicates that a family medicine department needs manpower and material resources to provide a full range of family medicine education, clinical services, and research plans [3]. Human resources such as mentorship, include faculty doctors and other personnel, who have sufficient time to teach and fully supervise trainees, develop courses, and conduct research. These faculty members usually include a combination of professionally trained family medicine and enthusiastic specialist physicians. A community-based family doctor may be hired as a full-time teacher, part-time clinical instructor, or mentor [3]. Existing literature describes mentorship as being vital to academic success. Thus, lack of quality mentorship has been identified as an impediment to a successful academic career, while having a mentor has been associated with increased career satisfaction, academic productivity, and a sense of community. However, despite these perceived benefits, less than half of the faculty members in the United States have a current mentor or are mentors themselves [19,20]. According to our literature review, mentorship does not exist between senior and junior faculty in the field of academic family medicine in Taiwan, except for medical students, interns, residents, and postgraduate programs [21]. Therefore, mentorship should be established in academic family medicine and other academic fields in Taiwan.

In our study, four of the ninety adjunct faculty members are community-based family physicians, while others are adjunct or full-time university-based family physicians offered clinical instruction or mentor work. Moreover, only twenty-one are full-time teachers in medical schools. Our research also indicates that most medical faculty members in Taiwan are adjunct faculty and do not work full-time. This may be because most of them are physicians in medical centres, performing clinical work mostly while being partially involved in research or teaching, which is consistent with a previous study [22]. A proactive approach to family medicine faculty retention begins with recruitment. Thus, more full-time faculty members can be retained in academic family medicine in Taiwan through improved compensation packages, benefits, faculty development, work–life balance, and appropriate protected time. Retention efforts are guided and modified through open and frequent communication between faculty members and administration. As other studies have suggested, the Family Medicine Educator Fellowship [23], a framework for providing family medicine skills training, a family medicine course curriculum for teaching skills training, and mentorship programs for family medicine should be established [24]. We also found that family medicine academic faculty members are more likely to continue in institutions, where they receive the opportunity to participate. Thus, if the teacher’s professional priorities align with the institution’s priorities, they are less likely to leave the academic institution as they feel valued by the agency and find it reliable [25].

Undergraduate family medicine teachers in medical schools should be specialists, or at least experienced trainees in education practice or family medicine. In Taiwan, all family medicine faculty members of medical schools are family medicine specialists, and we found that their subspecialties are diverse. Interestingly, one family medicine faculty member, in our research, was still under a dermatology residency training, despite already becoming a psychiatry and family medicine specialist. This also shows that family medicine is generalist and cross-disciplinary, which may have multiple subspecialties [26].

Additionally, all the the academic family medicine faculty in our study possessed an MD degree, with some possessing inter-disciplinary PhDs. This further indicates the frequently occurring cross-disciplinary nature of family medicine faculty.

In our study, the main subspecialties of academic family medicine teachers in Taiwan were geriatric medicine, and hospice palliative medicine. These fields address Taiwan’s transformation into an aging society and its long-term care needs since 2003 (older adults over 65 years old may account for 20.1% and 41.6% by 2025 and 2070, respectively) [5]. Therefore, the current family medicine education program should cover topics regarding undergraduate family medicine, gerontology, and hospice palliative medicine, and the concept, and education above this should not wait until medical students graduate and enter the clinic to learn. Moreover, the results also show that as the academic rank of faculty increases, their age gradually increases, and their seniority is longer. The average age of lecturers was younger, and professors were mostly older than the other ranks. This result is related to Taiwan’s current system for medical academic promotion and the need for instructors to serve as role models with years of practice and current knowledge [12].

Our research shows that most faculty members in the Department of Family Medicine are males (73.3% vs. 26.7%), and these results are similar to those in the United States in 2001 (62.7% vs. 37.3%) [27]. However, research on gender disparity in academic family medicine in North American medical schools in 2020 represents that females hold 46.11% of faculty positions and 41.5% of professor positions in 2020 [28]. Although our study did not collect longitudinal data, it shows that the proportion of female teachers in family medicine, who were promoted to professors was zero in 2019. Therefore, the proportion of female teachers being promoted to professors is lower than that of their male counterparts and needs to be improved in the academic family medicine workforce in Taiwan [29]. This result indicated that the gender distribution of the academic family medicine workforce in Taiwan is still years behind that in the United States. This might be associated with the continuously widespread male dominance over all levels of medical training in Taiwan [30,31], combined with the insufficient number of female occupying a leading position in the academic environment [27]. In addition, female teaching positions in Taiwan may be affected by Asian social traditions, such as the impact of childbirth, raising children, and family care on their careers, especially when facing academic career promotion issues [32,33].

The number of full-time faculty members in the family medicine department of medical schools in Taiwan increases with higher academic rank, while the number of adjunct faculty members decreases with lowered academic ranks. This finding represents that the full-time status of family medicine faculty might direct greater emphasis on their research and teaching work, which may result in a higher academic rank. Another challenge faced in medical education is that faculty often have limited time and resources for conducting research. Family medicine faculty in medical schools generally reflect the demographics of family medicine education programs and the activities of members of the Taiwan Academy of Family Physicians [34]. According to our findings, lecturers comprise the highest number of faculty members in medical schools by academic rank, followed by assistant professors and professors. This finding is quite similar to other countries [20], and it may be related to a rigorous promotion system, which makes promotion difficult. In addition, the number of medical faculty members in foreign medical schools, whether male or female, is higher for full-time than adjunct faculty members [10]. Conversely, in Taiwan, the total number of adjunct family medicine faculty members exceeds that of full-time medical faculty. It might be that the current family medicine education of medical schools in Taiwan has received little attention, so most faculty members remain adjunct instructors. In Taiwan, the number of female full-time faculty members is significantly lesser than female adjunct faculty members. However, the number of female adjunct faculty members is still lower than male adjunct faculty members. Female adjunct medical faculty members were most likely to be instructors. This finding may be related to the promotion systems of various medical schools in Taiwan. In particular, female teachers believe that the lack of support and recognition of the role of adjunct instructors will undermine their abilities due to associated “dejected” and “unvalued” attitudes [35]. Compared to males with children, females with children spend more time on childcare issues, encounter more difficulties at work, and work more on weekends. Furthermore, female faculty members are also more likely to choose non-traditional or part-time jobs to better balance the needs of students, their families, and work [36]. Some females believe that the lack of support and recognition for adjunct teachers weakens their ability to balance family needs and careers [27]. Governments or institutions that strive to help achieve this balance are more likely to retain female faculty. Therefore, our research results substantiated the current status of female family medicine teaching positions and the bottlenecks encountered in academic advancement. However, further investigation is needed to understand gender differences in medical teaching positions in medical schools in Taiwan.

A previous study showed that the shortage of primary care physicians in the United States is due to population growth and aging, physician retirement, and changing physician work patterns [37]. Similarly, our study shows that Taiwan faces the same problems concerning an aging population, physician retirement in family medicine, and a changing environment in the healthcare system. Thus, Taiwan can learn from the USA and address these issues, such as providing medical student internships, expanding pipeline programs, placing more emphasis on placing family doctors on medical school admissions committees, changing admission criteria to favour the selection of candidates with a higher likelihood of choosing family medicine, identifying new family physician mentors and role models, and rigorously evaluating the impact of curricular and extracurricular activities on students [38].

Family medicine is the area of medicine experiencing the most substantial obstacles of a shortage of teaching faculty. We hope that the Ministry of Education, and the health policy makers in the government institutions, will also allocate funds to develop and improve the standards of family medicine doctors. Training medical students to actualize the family medicine spirit is a future plan for medical schools. To create more opportunities for training and advanced studies, more family medicine role models in early medical school [39], such that the ordinary medical students can observe the importance of the doctor–patient relationship and the value of the accumulated knowledge about patients acquired over time, can be further promoted to family medicine type physicians after graduation, regardless of their specialty, and actively promote the academic family medicine workforce [40].

### 4.2. Study Limitations and Strengths

The present study had several limitations. First, no previous family medicine academic workforce study in Taiwan was available. Therefore, although our study showed family medicine faculty data in 2019, it cannot provide a complete picture of the overall development or identify trend in the family medicine academic workforce over the years. Second, our study only investigated the faculty members in the family medicine department of medical schools in Taiwan; thus, faculty members who were teaching family medicine at other non-medical schools or colleges, or hospitals were not included [41]. Lastly, our study shows the fundamental original data of the family medicine faculty in medical schools, without any information about job satisfaction and in-depth mentorship or academic track activity [20,42]. Therefore, further in-depth analysis and improvement of the family medicine academic workforce are needed.

Despite these limitations, our study demonstrates the start of and the first baseline research on the family medicine academic workforce in Taiwan. Additionally, this study highlights the corresponding research work around a large amount of actual survey data, which has certain social values and policy guidance. Understanding the current status of family medicine faculty in Taiwan medical schools will help improve the process of promoting family medicine academic faculty and resolve barriers in terms of gender, age, and geographic distribution, among others. Furthermore, it will help to formulate an effective family medicine education plan in Taiwan. The results of this study can also improve the quality of the healthcare system and social policy in Taiwan. 

### 4.3. Future Recommendations

The following recommendations can be drawn from our research findings:Efforts should be made to establish family medicine departments in medical schools in Taiwan to implement principles and practices of family medicine within the Taiwanese healthcare system.Family medicine should be embedded within the medical education curriculum at an early stage, to inform medical school students about primary care and family medicine practice and to be able to implement a holistic and comprehensive approach for health problems.Quality mentorship leads to a successful academic career and is associated with increased career satisfaction, academic productivity, and a sense of community; thus, it should be established in academic family medicine departments in Taiwan,Family medicine educator fellowship should be established, as a framework for providing family medicine skills training, a family medicine course curriculum for teaching skills training, and mentorship programs for family medicine in the future.Government or education institutions should strive to improve gender, age, regional, and other diversity among faculty members by providing promotional opportunities for diverse faculty in academic family medicine and overall academic medicine departments.

## 5. Conclusions

Our study provides the most complete census to date on academic physicians of family medicine in all medical schools in Taiwan. The results can provide educational leaders, policy managers, and decision makers with new information about the characteristics and current developments of the family medicine departments in Taiwan’s medical schools. It will help in the formulation of an effective mentorship within the academic departments of family medicine in Taiwan, while improving medical school family medicine education. Furthermore, these findings can provide a potential solution to the shortage and imbalanced between gender and regional distribution of family medicine faculty in medical schools of Taiwan to help academic family medicine and national health policy development in the future. Consequently, the significance of certain social values and policy guidance is established. 

## Figures and Tables

**Figure 1 ijerph-18-07182-f001:**
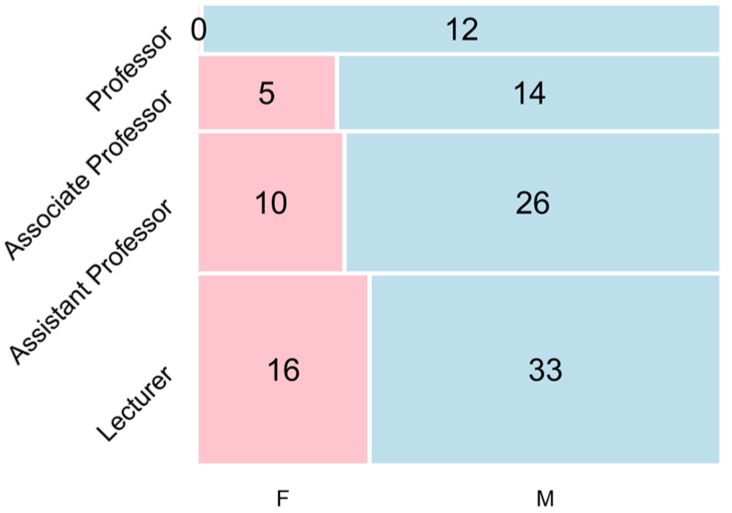
Gender and faculty rank of academic family medicine faculty.

**Figure 2 ijerph-18-07182-f002:**
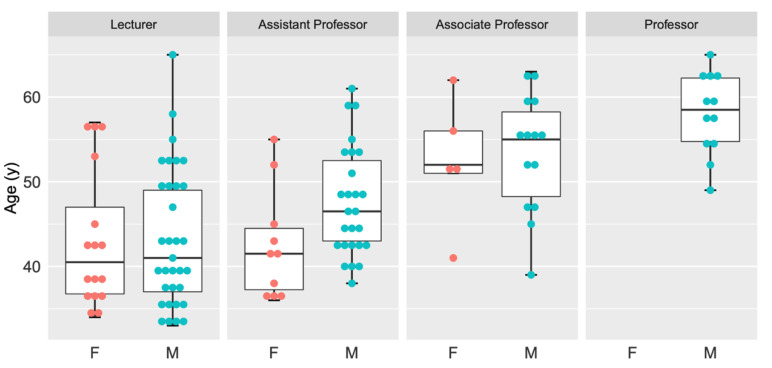
The age and gender distribution of family medicine academic faculty by rank.

**Table 1 ijerph-18-07182-t001:** Characteristics of medical schools in Taiwan (*n* = 13).

	Medical School Characteristics			
	Public		Private	
	*n*	%	*n*	%
With family medicine department				
North	3	23.1	1	7.7
Central	0	0	2	15.3
South	1	7.7	1	7.7
East	0	0	1	7.7
Total	4	30.8	5	38.4
Without family medicine department				
North	0	0	3	23.1
Central	0	0	0	0
South	0	0	1	7.7
East	0	0	0	0
Total	0	0	4	30.8

**Table 2 ijerph-18-07182-t002:** Distribution of family medicine academic physicians of medical schools in Taiwan (*n* = 13).

	Total Family Medicine Academic Physician in Taiwan (n = 116)			
	Public (*n* = 63, 54.3%)		Private (*n* = 53, 45.7%)	
	*n*	%	*n*	%
With a family medicine department (n = 99, 85.3%)	63	100	36	67.9
Without a family medicine department (n = 17, 14.7%)	0	0	17	32.1
North (n = 79, 68.1%)	58	92	21	39.6
Central (n = 14, 12.1%)	0	0	14	26.4
South (n = 13, 11.2%)	5	8	8	15.1
East (n = 10, 8.6%)	0	0	10	18.9

**Table 3 ijerph-18-07182-t003:** Characteristics of family medicine faculty in medical schools in Taiwan (*n* = 116).

Characteristic		*n*	%
Gender			
	Male	85	73.3
	Female	31	26.7
Age (y)			
	30–39	22	19.0
	40–49	40	34.5
	50–59	39	33.6
	60–69	15	12.9
Work status			
	Full-time	26	22.4
	Adjunct	90	77.6
Academic position			
	Professor	12	10.3
	Associate Professor	19	16.4
	Assistant Professor	36	31.0
	Lecturer	49	42.2
Faculty year			
	<10 years	94	81.1%
	10–20 years	20	17.2%
	>20 years	2	1.7%
Faculty degree			
	PhD	38	32.8
	Master	49	42.2
	Bachelor	29	0.25
Subspecialty			
	Gerontology and geriatrics medicine	55	47.4
	Hospice palliative medicine	53	45.7
	Environmental and occupational medicine	13	11.2
	Obesity medicine	11	9.5
	International travel medicine	5	4.3
	Adolescent medicine	3	2.6
	Osteoporosis medicine	2	1.7

**Table 4 ijerph-18-07182-t004:** Academic rank distribution by gender for all full-time family medicine physician faculty members in Taiwan medical schools in 2019.

Rank	All Faculty		Male Faculty		Female Faculty	
	*n*	%	*n*	%	*n*	%
Full-time faculty						
Professor	10	38. 5	10	50	0	0
Associate Professor	9	34.6	7	35	2	33.3
Assistant Professor	6	23.1	3	15	3	50
Lecturer	1	3.8	0	0	1	16.7
Total	26	100	20	100	6	100
Adjunct time faculty						
Professor	2	2.2	2	3.0	0	0
Associate Professor	10	11.1	7	10.8	3	12
Assistant Professor	30	33.3	23	35.4	7	28
Lecturer	48	53.4	33	50.8	15	60
Total	90	100	65	100	25	100

## Data Availability

Not applicable.

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
