# Peer review of "Family Medicine Academic Workforce of Medical Schools in Taiwan: A Nationwide Survey"

_ijerph, 2021, doi:10.3390/ijerph18137182_

Round 1

Reviewer 1 Report

The paper carried out corresponding research work around a large amount of actual survey data, which has certain social value and policy guidance significance. The results of the thesis help to formulate an effective family medicine education plan. However, the paper has some normative deficiencies that need to be improved, and the research depth is slightly insufficient. It is recommended that more in-depth data mining and analysis work be carried out in the future.

Author Response

Please see the attachment. Thank you very much :)

Reviewer 2 Report

This manuscript is only a basic report of FM academic workforce of medical schools in Taiwan. What are the meanings in clinical medicine, education/training program or national health policy ?

Perhaps the authors could analyze a 10-years trend of FM academic workforce and FM clinical physicians compared to the change of medical utilization (i.e. direct expenditure paid by the Taiwan's NHI) in Taiwan.

Reviewer 3 Report

This article provides a thorough overview of the medical faculty in Taiwan, and the results are clear and well-presented, and this paper was well-written. I have several suggestions to improve the narrative of this paper and emphasise the importance of the findings. 

Introduction

line 36-37 - Having a definition of family medicine is useful for the reader, however this definition is hard to understand 

line 62-65 - This is good information to justify the importance of family medicine doctors. I think more is needed here to emphasise the role of family medicine doctors as primary caregivers. 

line 67 - The justification for looking at the faculty of family medicine is not clear. More information is needed to justify why it is important to understand the characteristics of the faculty and what benefits this information would have. 

Methods

line 95 - more description of data analysis is needed. What analyses were done and what was the aim? i.e. were they descriptive analyses to summarise key variables? 

Results - overall the results section was informative and the key variables were clear. 

Discussion - The discussion is clear and well-written. Overall there could be more contextual information, for example examples from research in other countries and a comparison to the medical faculty found in this study. Areas for future research could also be expanded. 

Round 2

Reviewer 2 Report

Please focus on some possibly unique findings in Taiwan as below, that should be compared with other countries :

  1. maybe fewer faculty members in Taiwan’s medical schools (only 2.2% of specialists in FM)
  2. maybe lower educational level of these faculty members (only 32.8% with PhD degree)
  3. maybe lower academic level of these faculty members (the mostly, lecturers(42.2%))

Based on the above, please provide some recommendations to the health policy-makers in Taiwan.
